# MIL-88-Derived N and S Co-Doped Carbon Materials with Supplemental FeS$_x$ to Enhance the Oxygen Reduction Reaction Performance

Yu Liu [1,2,†], Yinghao Xu [2,3,†], He Wang [2,3], Jia Zhang [4], Haiyan Zhao [5], Li Chen [6], Ling Xu [1,*], Yan Xie [2,*] and Jiahui Huang [2]

1   College of Chemistry and Materials Science, Inner Mongolia Minzu University, Tongliao 028000, China; yuliu@dicp.ac.cn
2   Dalian National Laboratory for Clean Energy, Dalian Institute of Chemical Physics, Chinese Academy of Sciences, Zhongshan Road 457, Dalian 116023, China; yinghao.xu@uzh.ch (Y.X.); hw122@ic.ac.uk (H.W.); jiahuihuang@dicp.ac.cn (J.H.)
3   Dalian Leicester Institute, Dalian University of Technology, Dalian 116024, China
4   State Key Laboratory of Catalysis, Dalian Institute of Chemical Physics, Chinese Academy of Sciences, Zhongshan Road 457, Dalian 116023, China; zhangjia610@dicp.ac.cn
5   Liaoning Key Laboratory of Plasma Technology, School of Physics and Materials Engineering, Dalian Minzu University, Dalian 116600, China; zhaohaiyan@dlnu.edu.cn
6   Shanghai Key Laboratory of Green Chemistry and Chemical Processes, School of Chemistry and Molecular Engineering, East China Normal University, Shanghai 200062, China; chenli@chem.ecnu.edu.cn
*   Correspondence: tlxuling1979@163.com (L.X.); yanxie@dicp.ac.cn (Y.X.)
†   These authors contributed equally to this work.

**Abstract:** To overcome the drawbacks of the single N-doped carbon materials, the further development of dual-heteroatoms (N and S) co-doped electrocatalysts is highly anticipated. Herein, N, S-doping and Fe-based carbon materials were synthesized by pyrolyzing a metal–organic framework (MIL-88) with the addition of N-/N, and S-containing ligands (chitosan and L-Cysteine) in the case of iron salt. The resulting electrocatalyst heat-treated at 850 °C (FeNSC-850) displays superior oxygen reduction reaction (ORR) performances to MIL-88-850, with an overall electron transfer number of 3.97 and a minor yield of HO$_2^-$ (<2.6%). In addition to the comparable activity to commercial Pt/C in catalyzing the ORR in alkaline solution, the FeNSC-850 also shows higher stability, with a slight decline in half-wave potential ($\Delta E_{1/2}$ = 15 mV) after 5000-cycle scanning of cyclic voltammetry. In view of the multiple Fe-based active sites, the additional S doping within FeNSC-850 creates more FeS$_x$ active sites for boosting the ORR performances in alkaline solution.

**Keywords:** heteroatom doping; FeNSC electrocatalyst; oxygen reduction reaction

## 1. Introduction

As next-generation energy conversion and storage devices, fuel cells (FCs) are becoming appealing owing to their high energy density and environmental friendliness [1,2]. The overall efficiency of FCs is determined by the progress of the oxygen reduction reaction (ORR). To date, Pt was commonly used as the best ORR electrocatalysts, whereas the drawbacks of Pt, like high-cost and low-reserve, are topical for a large scale for FCs. Thus, it is inevitable to exploit the alternatives to Pt for FCs. The non-noble metal electrocatalysts (NNMEs), especially for N-doping and Fe-based carbon materials, hitherto have attracted much attention because of their excellent electrochemical catalytic properties for ORR in both alkaline and acidic solution [3,4], while, the N dopant with strong electronegativity will undesirably perturb the electronic properties of metal-based active sites and increase the free energy for the adsorption and activation of the intermediates towards the ORR, restricting the promotion of the intrinsic nature of NNMEs for ORR [5–7]. Thus, a secondary heteroatom dopant, such as S or P, is acceptable to optimize the microenvironment of metal

active centers [8]. It has been noted that N, S co-doping is superior to N, P or N, B co-doping in Fe-based NNMEs towards the ORR [9].

Recently, FeNSC electrocatalysts with a second heteroatom of S exhibit superb ORR performances due to S that can induce charge enrichment in N and Fe for optimal $O_2$ binding and fast electron transfer [10]. A series of N, S co-doping and transition metal-based carbon materials has been investigated on both experiments and theoretics for the ORR. Examples include, but are not limited to N, S co-doping and Fe-based porous carbon nanosheets [11–14]; N, S co-doping and Co/Fe-based carbon nanotubes [15,16]; N, S co-doping and Fe-based hierarchically porous carbon nanorods [5]; N, S co-doping and Co-based hollow carbon spheres [8]; N, S co-doping and Fe-based hierarchically porous honeycomb [17]; and N, S co-doping and Fe-based blocks [18]. All these electrocatalysts demonstrate considerable ORR performances in both alkaline and acidic solution.

In particular, metal–organic frameworks (MOFs) have been considered as ideal substrates due to facile synthetic pathways, their large porosity, and structural versatility after pyrolysis [19–21]. Among them, the MIL-88 framework could be chosen as a sacrificial template for a high graphitized structure [22,23]. As reported, iron carbides possess good ORR performances in alkaline solution [24], while the $Fe_3C$ derived from the carbonization of MIL-88 shows limited ORR performances, which can be improved by the exposure of dense Fe-based active sites through modifications on MIL-88 [23]. Herein, to address this conventional drawback of MIL-88, we innovatively utilized the synergistic effect of N and S dopants by annealing MIL-88 and additional Fe salt with the assistance of chitosan and L-Cysteine (L-Cy). Especially, chitosan is a low-cost and abundant biomass-derived ligand with strong coordination with metal, and L-Cy is a N, S-containing amino acid that is commonly used to introduce S in electrocatalysts for the ORR [25]. Consequently, the simultaneous introduction of N and S doping in carbon skeletons improves the stability of Fe by the formation of $FeN_x$, $Fe_xC$, and $FeS_x$ to boost the ORR performances in alkaline solution. Combining the porosity structure; N, S-dopants; and multiple Fe-based active species, the resulting electrocatalyst of FeNSC-850 shows excellent ORR performances via a direct four-electron transfer in 0.1 M KOH solution.

## 2. Results and Discussion

As depicted in Scheme 1, the N and S co-doping and iron-based carbon material was prepared by carbonizing the mixture of MIL-88, Fe salt, chitosan, and L-Cy at 850 °C. In this synthetic system, chitosan and L-cy were introduced as N-/N, S-containing biochemical precursors. During the pyrolysis step, N and S sources could bind with Fe to create more active sites, such as iron-based nitrides, carbides, and sulfides within carbon materials. Consequently, the porous N, S-co-doping and iron-based carbon materials, denoted as FeNSC-850, were designed and synthesized with excellent catalytic activities towards ORR in alkaline solution.

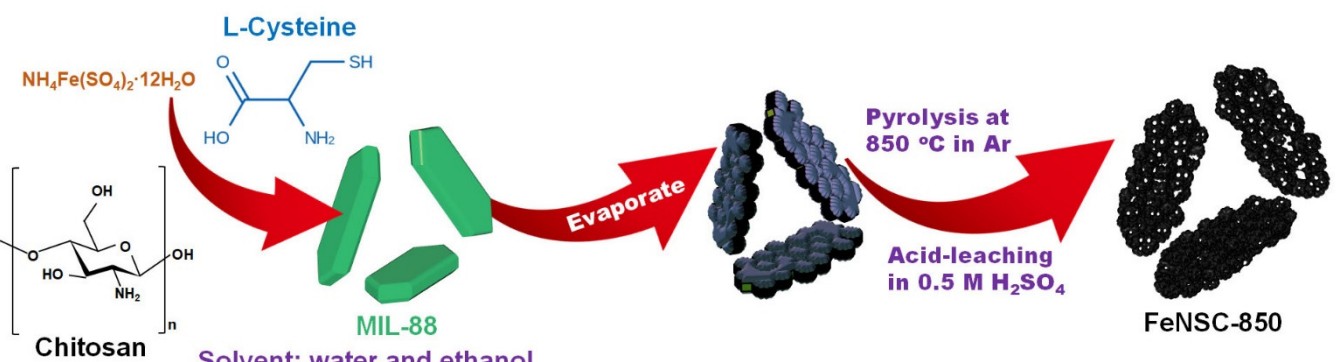

**Scheme 1.** Schematic illustration for the synthesis of FeNSC-850.

The morphology of MIL-88-850 in Figure S1a,b (in the Supplementary Materials) and FeNSC-850 (Figure 1a,b) are characterized by the scanning electron microscopy (SEM). After the carbonization of MIL-88 at 850 °C, a spindle-like structure of MIL-88-850 is observed in Figure S1a (in the Supplementary Materials). The enlarged SEM image in Figure S1b (in the Supplementary Materials) shows multiple pores on the surface of spindle-like structures after the pyrolysis process. With the addition of doping materials on MIL-88 in Figure 1a, the morphology of FeNSC-850 changed comparable with the starting templet of MIL-88. FeNSC-850 shows a hierarchical porous structure, including macro-, meso-, and micropores in the enlarged SEM image in Figure 1b. The energy-dispersive X-ray spectroscopy (EDX) pattern in Figure S2 (in the Supplementary Materials) indicates the presence of N, C, S, and Fe in FeNSC-850. As detected in Figure 1c, a high angle annular dark-field scanning TEM (HAADF-STEM) image and the element mapping of FeNSC-850 verify that C, N, and O are uniformly distributed throughout the whole structure, indicating the successful N doping within carbon matrixes. The overlap of S and C in element mapping clearly confirms the doping of S in carbon matrices. Simultaneously, the mapping results of Fe and S are superimposed, which indicates the sulfidation of Fe species encapsulated within carbon matrixes.

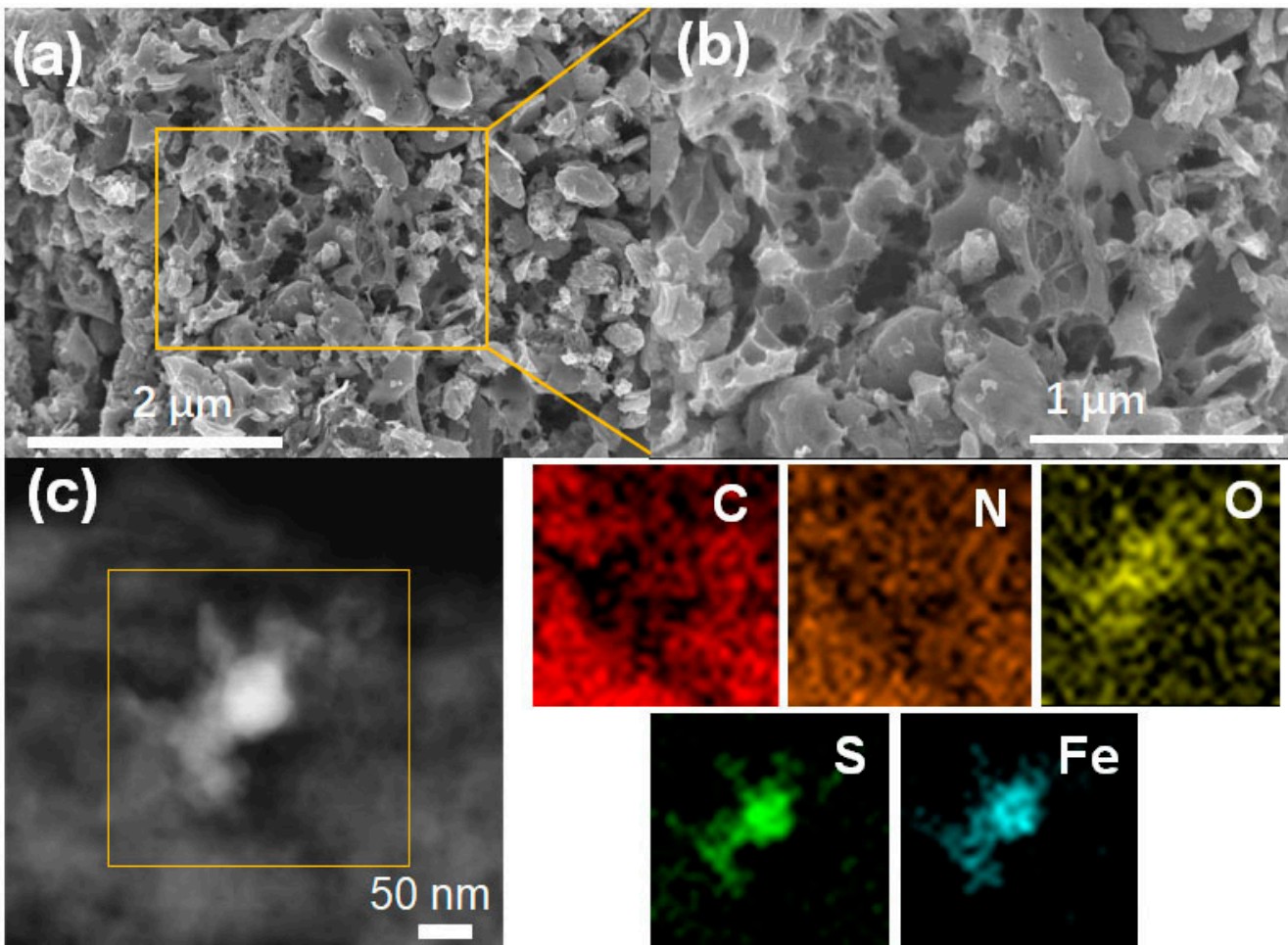

**Figure 1.** SEM images of (**a**) and (**b**) FeNSC-850; (**c**) the HAADF-STEM image of FeNSC-850 and its corresponding element mapping of C, N, O, S, and Fe.

The nature of samples was further analyzed by X-ray diffraction (XRD), as shown in Figure 2a and Figure S3 (in the Supplementary Materials). It is turned out that the peaks of as-prepared electrocatalysts appearing around 25° and 44° are indexed to (002) and (101)/(100) planes of graphite carbon [26]. A sharper and narrower peak around

26.4° of FeNSC-850 can be ascribed as $Fe_3C$ (PDF #03−0411) [27]. Notably, the apparent diffraction peaks of MIL-88-850 at 42.9, 43.7, 44.6, 44.9, 45.9, 48.6, and 49.1° correspond to (211), (102), (220), (031), (112), (131), and (221) planes of $Fe_3C$ (PDF #35−0772) in Figure S3a (in the Supplementary Materials) [28]. In comparison, the well-defined diffraction peaks of FeNSC-850 around 33.1, 37.1, 40.8, 47.5, 50.3, and 56.3° probably arise from the (200), (210), (211), (220), (221), and (311) planes of $FeS_2$ (PDF #42−1340) [29]. Moreover, the XRD patterns at around 20.8° and 50.3° could be linked to the (101) and (103) planes of FeS (PDF #76−0965). Accordingly, combined with the element-mapping results of Fe and S (Figure 1c), it is clear that the addition of L-Cy promotes the conversion of Fe ions to $FeS_x$. The appearance of $FeS_2$ in electrocatalysts not only promotes the smooth transfer of electrons to active sites but also strengthens the conductivity of electrocatalysts [30,31]. The Raman spectra of MIL-88-850 and FeNSC-850 in Figure S4 (in the Supplementary Materials) show two sharp peaks at around 1350 and 1600 $cm^{-1}$, corresponding to defects (D band) and the planar motion of $sp^2$-carbon atoms in graphite carbon (G band) [32]. The ratio of the D and G band ($I_D/I_G$) characterizes the values on MIL-88-850 ($I_D/I_G = 0.899$) and FeNSC-850 ($I_D/I_G = 0.854$), indicating the higher carbon graphitization of FeNSC-850.

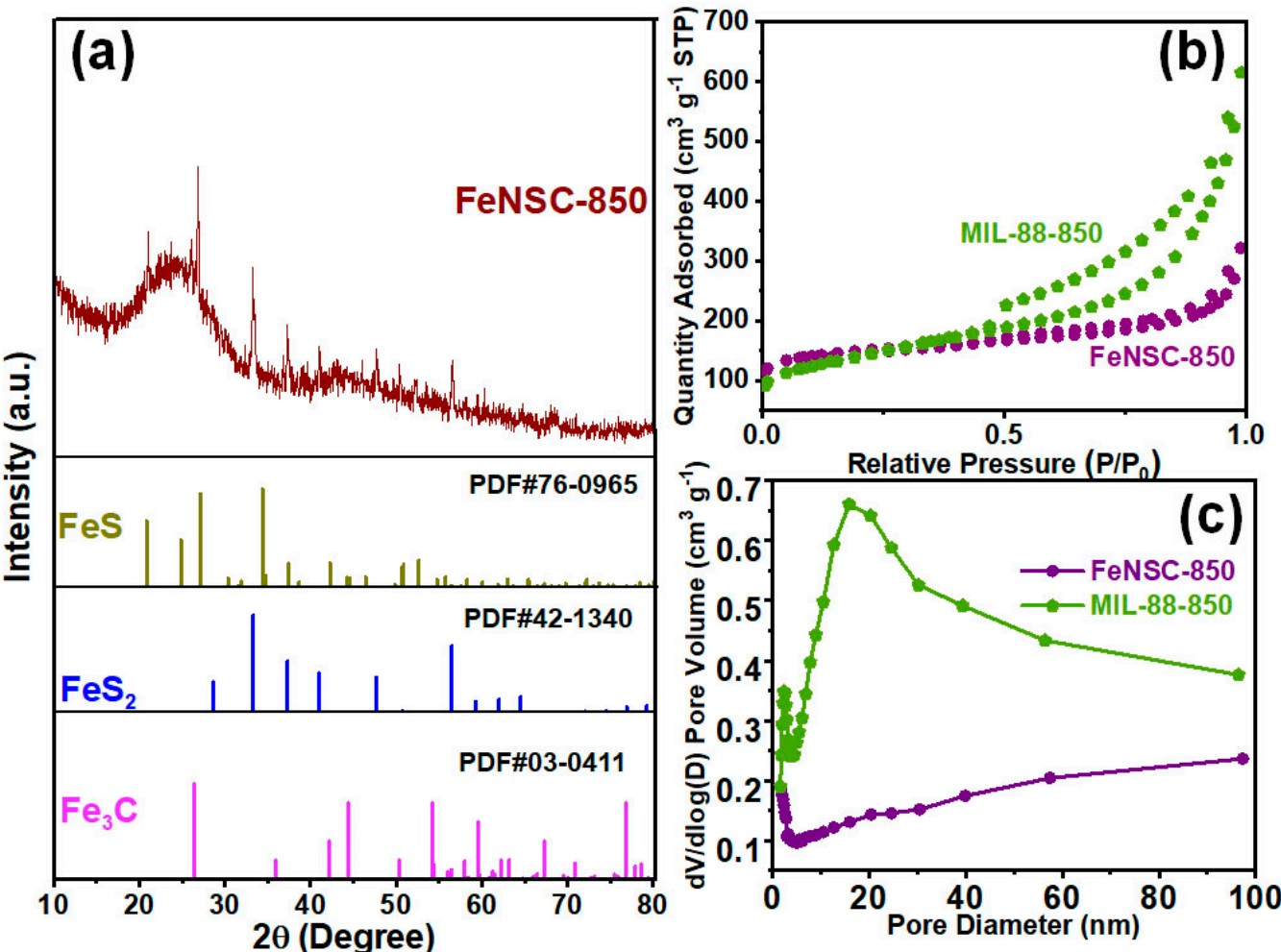

**Figure 2.** Physical characterizations of structure and composition of electrocatalysts: (**a**) X-ray diffraction pattern of FeNSC-850 accompanied by the standard XRD patterns of related substances of FeS (PDF#76−0965), $FeS_2$ (PDF#42−1340), and $Fe_3C$ (PDF#03−0411); (**b**) $N_2$ adsorption–desorption isotherms and (**c**) corresponding pore size distributions of FeNSC-850 and MIL-88-850.

The porosity of as-synthesized electrocatalysts was characterized by $N_2$ adsorption-desorption analysis in Figure 2b,c. Both FeNSC-850 and MIL-88-850 possess a type-IV

isotherm in Figure 2b. This suggests the existence of mesopores in the electrocatalysts, which is detected by SEM in Figure 1 and confirmed by pore size distribution shown in Figure 2c. The Brunauer–Emmett–Teller (BET) surface area ($S_{BET}$ = 476 m$^2$ g$^{-1}$) of FeNSC-850 is similar to that of MIL-88-850 ($S_{BET}$ = 483 m$^2$ g$^{-1}$). The microporous surface area ($S_{Micro}$ = 291 m$^2$ g$^{-1}$) of FeNSC-850 with a volume of micropore ($V_{Micro}$) of 0.15 cm$^3$ g$^{-1}$ by t-Plot is significantly larger than that of MIL-88-850 ($S_{Micro}$ = 97 m$^2$ g$^{-1}$, $V_{Micro}$ = 0.05 cm$^3$ g$^{-1}$ by t-Plot), revealing more accommodation for hosting a high density of Fe-based active species for the enhancement of ORR activities. The release of NH$_3$ and H$_2$S from L-Cy at high pyrolysis temperature leads to a larger $S_{Micro}$ and $V_{Micro}$, which is believed to be a major benefit for oxygen diffusion and mass transport to boost catalytic activities towards ORR [33]. Additionally, the formation of FeN$_x$ moieties has been found to be more favorable in micropores in carbon matrixes. As shown in Figure 2c, the corresponding pore size of MIL-88-850 centers around 2.45 and 16.1 nm. In contrast, the size of the partial pore of FeNSC-850 decreases to around 1.74 nm, and the others are larger pores with a very wide pore size distribution. The observed meso- and macropores are constructed by removing the NPs after the acid-leaching step. In conclusion, the carbonization of N-/N, S-containing ligands can enhance the porosity of FeNSC-850 and produce a larger proportion of micropores for better ORR enhancement.

X-ray photoelectron spectroscopy (XPS) analyses were subsequently employed to characterize the surface electronic states of as-prepared electrocatalysts. The survey spectra of MIL-88-850 and FeNSC-850 were shown in Figure S5 (in the Supplementary Materials), indicting the presence of N, C, and O for MIL-88-850 and N, C, O, and S for FeNSC-850. There is not an obvious signal of Fe in both electrocatalysts, possibly due to the limited detect of XPS for Fe NPs encapsulated within thicker carbon layers. Specifically, the high-resolution C 1s spectrum of FeNSC-850 in Figure 3a could be deconvoluted into four peaks at 284.8, 285.8, 287.0, and 290.4 eV, respectively, corresponding to C=C-C (sp$^2$ graphitic carbon), C=N/C-O/C-S, C-N/C-O-C, and O-C=O, indicating that the heteroatoms (O, N, and S in our case) have been doped in the carbon skeleton [34]. The N 1s XPS spectra in Figure 3b could be categorized into five peaks after fitting at 398.4, 399.3, 401.0, 402.3, and 406.0 eV, respectively, correlated to pyridinic N, Fe-N, pyrrolic N, graphitic N, and oxidized N. It is known that pyridinic N, Fe-N, and graphitic N could contribute to promote the ORR in an alkaline solution. Particularly, graphitic N is beneficial to raise the conductivity of materials by influencing the electronic and geometric structures of carbon [35]. Since the content of the above-mentioned five types of N varies with pyrolysis temperature, the five nitrogen contents (relative to C) in different FeNSCs synthesized at four temperatures and MIL-88-850 are illustrated in Figure 3c and Figure S6 (in the Supplementary Materials). Notably, the graphitic N in FeNSC-850 is higher than samples pyrolyzed at other temperatures. In particular, the N content of FeNSC-850 (relative to C) is about 3.6 at%, which is higher than that of MIL-88-850 (1.8 at%). It confirms that the additional N-containing ligands indeed increase the N-dopant content of electrocatalysts. The high-resolution S 2p spectra of FeNSC-850 in Figure 3d shows that the strong peaks at 164.1 and 165.3 eV correspond to S 2p$_{3/2}$ and S 2p$_{1/2}$ of thiophene-like structures (C-S-C), respectively. The two small peaks at the binding energy of 168.5 and 169.7 eV arise from the oxidized S species, namely, -C-SO$_x$-C bonds [1]. Introducing S plays a positive role in the Fe-based microenvironment that promotes the interactions between Fe active sites and oxygenated reactive intermediates in the ORR process [14]. More precisely, the observed C-S-C species are reported to contribute to more effective facilitation towards the ORR than -SO$_x$. [36]. The oxidation state of the surface iron is interpreted from the spectra characteristic peaks in Figure 3e. As a result, the peaks centered at around 711.3 and 714.1 eV are from Fe$^{2+}$ and Fe$^{3+}$, respectively. It is found that the positive shift of the high binding energy from 710 to 711 eV is due to heteroatoms (N and S in our case) that decrease the charge density on Fe [33]. Obviously, the binding energy at around 707.3 eV is attributed to FeS$_2$ [37], which is in line with the result of XRD patterns in Figure 2a. Moreover, the Fe content of MIL-88-850 is determined by inductively coupled

plasma optical emission spectroscopy (ICP-OES) as 9.34 wt%. Additionally, the Fe content of FeNSC-850 is determined by ICP as 3.37 wt%. Obviously, FeNSC-850 lost some Fe content during the high-temperature pyrolysis and acid-leaching. As shown in the SEM images in Figures 1 and S1 (in the Supplementary Materials), the structure of MIL-88-850 is maintained, while some of the structure of FeNSC-850 collapsed; thus, some Fe-based NPs leached out and Fe content decreased.

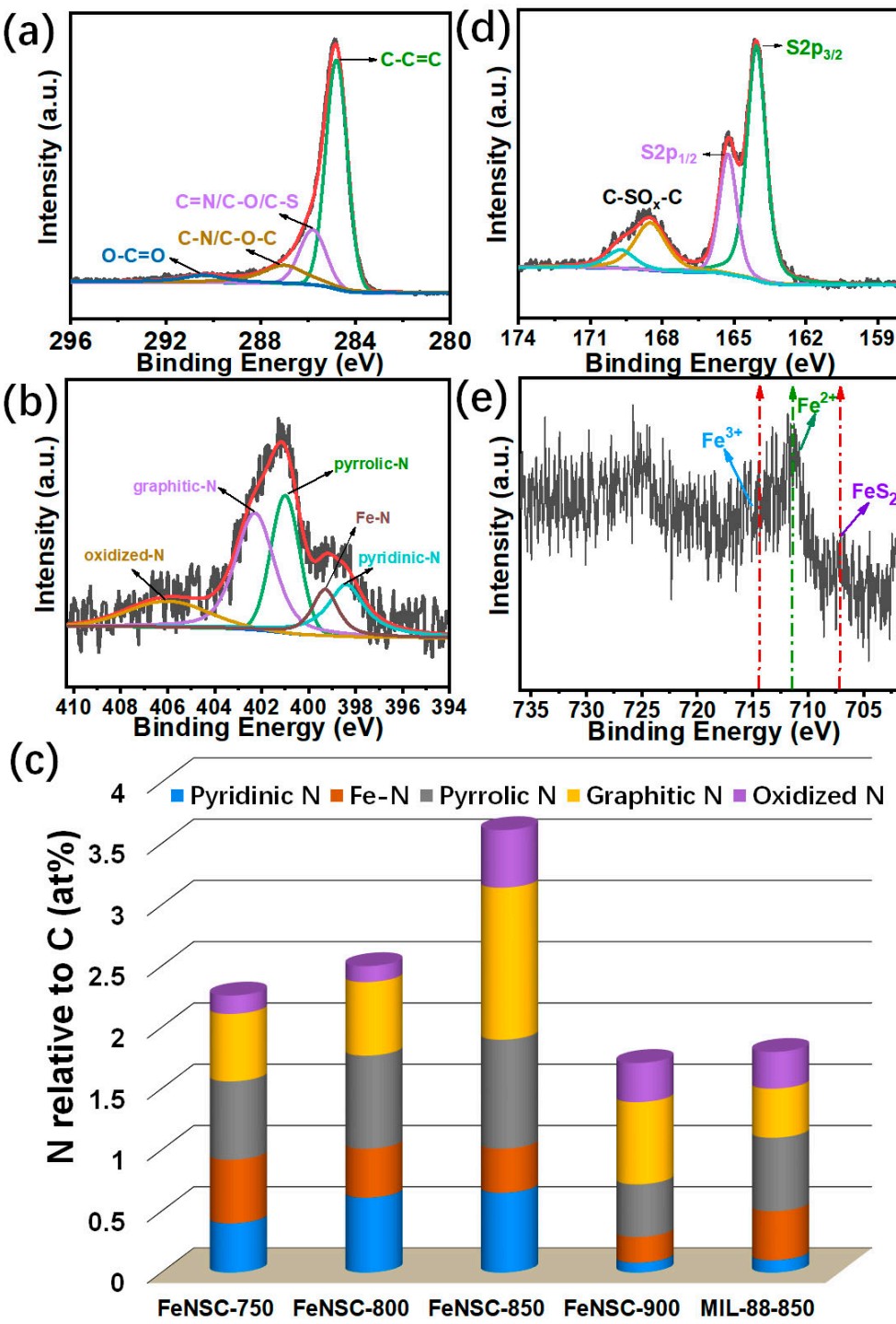

**Figure 3.** XPS (**a**) C 1s and (**b**) N 1s spectra of FeNSC-850; and (**c**) the N content relative to C in electrocatalysts of FeNSC-T and MIL-88-850, XPS (**d**) S 2p, and (**e**) Fe 2p spectra of FeNSC-850.

The $^{57}$Fe Mössbauer spectroscopy is further used to identify the chemical environment of iron in the electrocatalysts (Figure 4a,b and Table 1). Distinctly, the spectrum of MIL-88-850 in Figure 4a is composed of two typical components, including a doublet ascribed to $Fe^{2+}N_4$ (D1, 23.34%) in the hexagonal symmetry site and a sextet (S1) assigned to $Fe_3C$ (S1, 76.66%). The spectrum of FeNSC-850 in Figure 4b is composed of six sextets and two doublets. The fitted Mössbauer parameters in Table 1 indicate that a sextet is assigned to $Fe_xC$ (S2, 11.22%). Three sextets, including S3, S4, and S5, correspond to the various iron sulfides denoted as $FeS_x$ (61.73%). Two doublets of D2 and D3 are assigned to $FeS_2$ (22.05%) and $Fe^{2+}N_4$ (5.01%) in the tetrahedral symmetry site, respectively. In contrast to $FeN_4$ (hexagonal symmetry site) in MIL-88-850, the addition of N-containing ligands plays an important role in controlling the insertion of N into the lattice of Fe matrixes such as $FeN_4$ (tetrahedral symmetry site) for different folds as the coordination formation in FeNSC-850. Moreover, the S-containing ligand, namely, L-Cy in FeNSC-850 controls the insertion of S into the lattice of Fe matrixes for supplemental $FeS_x$, which may alter the surface electron configuration for the adsorption/activation of $O_2$ and further provide networks for efficient electron transfer [1]. As a result, $FeS_x$ is regarded to be the main Fe-based active sites in FeNSC-850.

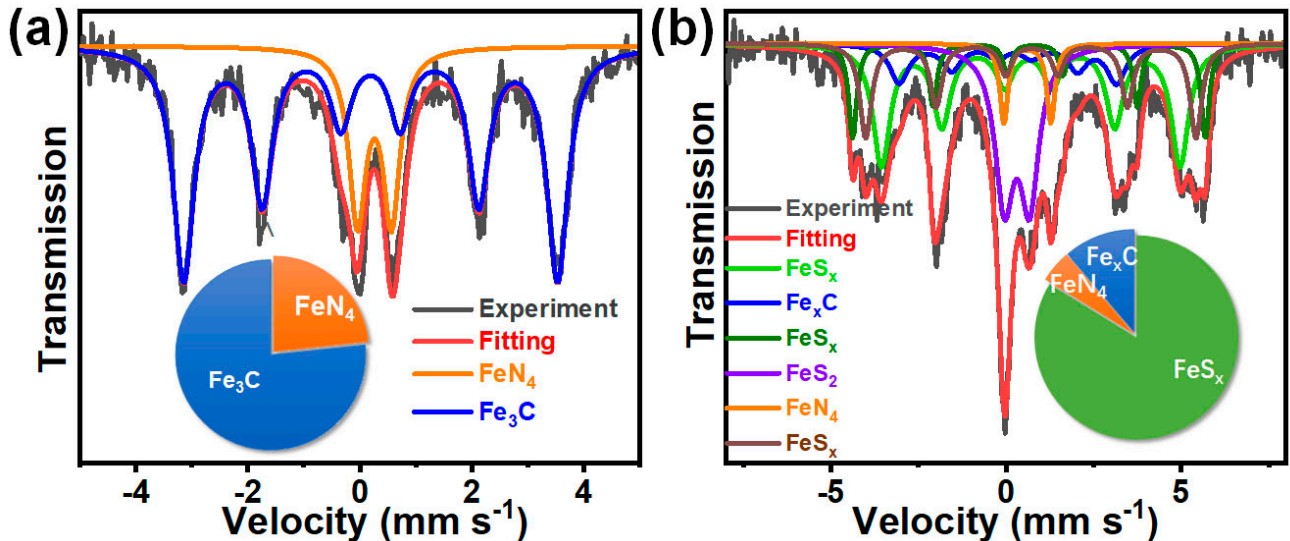

**Figure 4.** Mössbauer spectra of (**a**) MIL-88-850 and (**b**) FeNSC-850, with the assignment of Mössbauer sites to iron species in the relative absorption area.

**Table 1.** The Mössbauer spectra parameters of MIL-88-850 and FeNSC-850.

| Sample | | IS [a] $(mm\,s^{-1})$ | QS [b] $(mm\,s^{-1})$ | LW [c] $(mm\,s^{-1})$ | Assignment | Ref. |
|---|---|---|---|---|---|---|
| MIL-88-850 | Doublet (1) | 0.26 | 0.61 | 0.37 | $Fe^{2+}N_4$ (hexagonal symmetry) | [38] |
| | Sextet (1) | 0.19 | 0.0028 | 0.45 | $Fe_3C$ | [38] |
| FeNSC-850 | Doublet (2) | 0.31 | 0.72 | 0.65 | $FeS_2$ | [38] |
| | Doublet (3) | 0.61 | 1.34 | 0.28 | $Fe^{2+}N_4$ (tetrahedral symmetry) | [39] |
| | Sextet (2) | 0.15 | −0.19 | 0.64 | $Fe_xC$ | [40] |
| | Sextet (3) | 0.67 | 0.071 | 0.60 | | |
| | Sextet (4) | 0.74 | −0.17 | 0.28 | $FeS_x$ | [38,41] |
| | Sextet (5) | 0.72 | −0.028 | 0.40 | | |

[a] IS isomer shift. [b] QS is quadrupole splitting. [c] LW is line width.

Generally, highly efficient NNMEs retained at moderating pyrolysis temperature are due to the balanced properties as for the graphitic structure, the surface area, and the obtained active species, such as C-S, C-N, Fe-N, and Fe-S. [14] The electrochemical study

of FeNSCs obtained from various pyrolysis temperatures (FeNSC-T) in Figure 5a was conducted in 0.1 M KOH solution at a rotation rate of 1600 rpm. For clarity, the potentials of electrocatalysts in Figure 5a are summarized and contrasted in Figure 5b at a representative current density of $-2.5$ mA cm$^{-2}$. The ORR performance of FeNSC-850 is enhanced relative to those of FeNSC-750 and FeNSC-800. Remarkably, a plummet of activity is observed from FeNSC-900, which can be explained by the decrease of N at% (N relative to C, 1.7 at%) in Figure 3c. Further treatment with a higher temperature probably wouldresult in lower yield of electrocatalyst. Thus, we did not try any other higher temperature over 900 °C for heating the samples. Consequently, FeNSC-850 displays the best ORR performances in terms of the onset potential ($E_{onset}$ = 0.89 V vs. RHE) and the half-wave potential ($E_{1/2}$ = 0.82 V vs. RHE) among FeNSC-T electrocatalysts, which is comparable to those of the commercial Pt/C with different Pt loadings (10 µg$_{Pt}$ cm$^{-2}$: $E_{onset}$ = 0.92 V and $E_{1/2}$ = 0.84 V vs. RHE; 20 µg$_{Pt}$ cm$^{-2}$: $E_{onset}$ = 0.94 V and $E_{1/2}$ = 0.85 V vs. RHE).

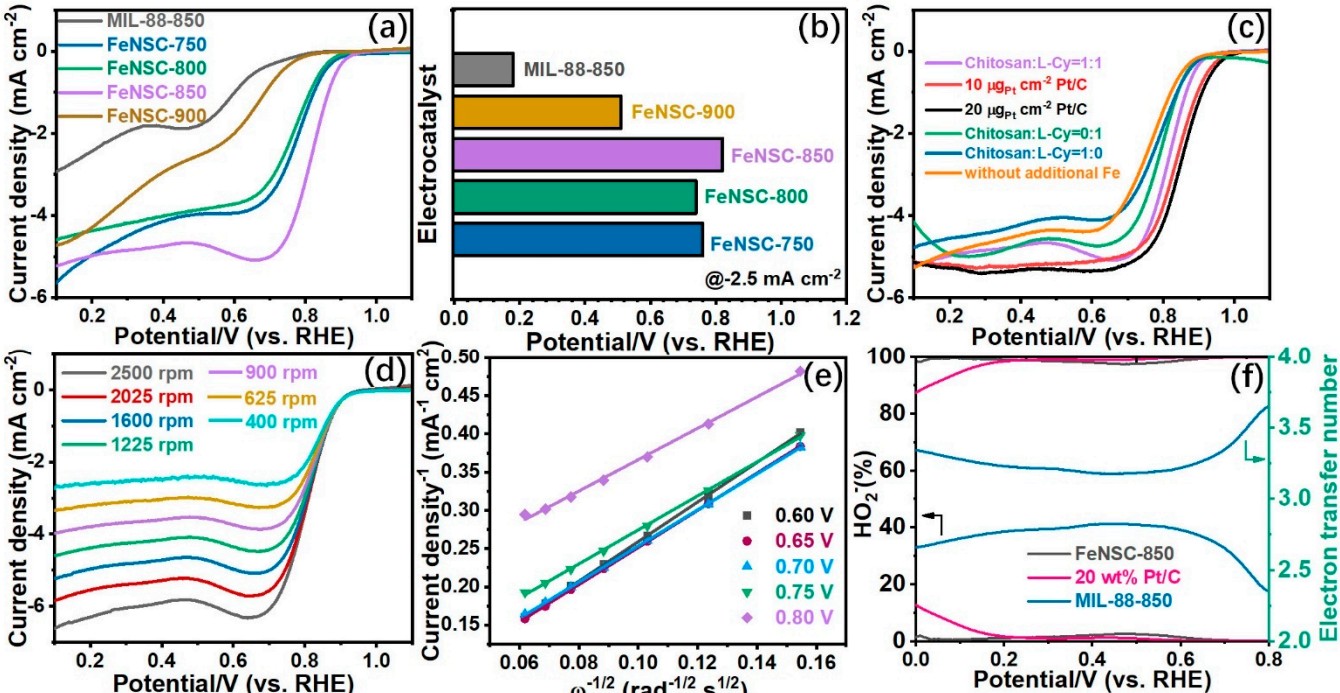

**Figure 5.** (**a**) Linear sweep voltammogram (LSV) curves of FeNSCs pyrolyzed with various heating-temperature from 750 to 900 °C and MIL-88 carbonized at 850 °C at 1600 rpm; (**b**) potentials of FeNSC-T (T = 750 − 900 °C) and MIL-88-850 at −2.5 mA cm$^{-2}$ based on the LSV data from (**a**); (**c**) LSV curves of as-prepared electrocatalysts with different mass ratio of chitosan and L-Cy, the electrocatalyst without additional Fe salt with mass ratio of chitosan and L-Cy as 1:1, and commercial Pt/C at loading of 10 and 20 µg$_{Pt}$ cm$^{-2}$ with a scanning rate of 5 mV s$^{-1}$; (**d**) LSV curves of FeNSC-850 with various rotation rates from 400 to 2500 rpm; (**e**) Koutecky-Levich (K-L) plots of FeNSC-850 based on the data from (**d**); and (**f**) HO$_2^-$% and electron transfer number of FeNSC-850, MIL-88-850, and 20 wt% Pt/C with Pt loading of 20 µg$_{Pt}$ cm$^{-2}$ in 0.1 M KOH electrolyte.

The addition of chitosan and L-Cy introduces more N and S elements within carbon matrices during high-temperature pyrolysis process. To investigate the aid of Fe salt, chitosan, and L-Cy in electrocatalysis for ORR, three groups of electrocatalysts without the addition of either Fe salt, chitosan, or L-Cy were synthesized in parallel with FeNSC-850. As shown in Figure 5c, all three as-prepared materials (where chitosan:L-Cy = 0:1, chitosan:L-Cy = 1:0, and the electrocatalyst without additional Fe salt with mass ratio of chitosan and L-Cy as 1:1) apparently present reduced activity compared to FeNSC-850 (where chitosan:L-Cy = 1:1), which confirms the importance of the synergistic effect of N and S elements within carbon matrixes at 850 °C and the necessity of Fe by coordination with N or S as the high-

density of active sites for boosting the ORR performances. This confirms that chitosan and L-Cy have a synergistic effect for preparing highly efficient electrocatalysts by increasing the active sites for ORR. According to the reported studies, doping electrocatalysts solely with sulfur actually would dampen their activity towards ORR. S-doping can, however, induce defects in the carbon skeleton and enrich the charge on adjacent Fe and N, thereby accumulating more electrons on adsorbed $O_2$ and facilitating the reductive release of *OH to enhance the ORR performance [42,43]. Thus, chitosan plays an important role in increasing the content of N dopants in electrocatalysts. Afterwards, supplemental $FeS_x$ active sites provided by the L-Cy could act as a booster for ORR. Generally, the superior ORR activity of FeNSC-850 is attributed to the co-function of Fe-based active sites and N, S co-doped carbon matrixes [44].

The LSV curves of FeNSC-850 in Figure 5d demonstrate that the increasing limited diffusion current densities increase along with the increasing rotation rates ranging from 400 to 2500 rpm. Accordingly, the Koutecky–Levich (K-L) plots of FeNSC-850 in Figure 5e derived from the data in Figure 5d suggest that the electron transfer number (n) of FeNSC-850 is calculated to be 3.92 at potentials from 0.6 to 0.8 V (vs. RHE). To further investigate the electron transfer number and $HO_2^-$ intermediate, rotating ring-disk electrode (RRDE) measurements were carried out with FeNSC-850, MIL-88-850, and 20 wt% commercial Pt/C (Figure 5f) in $O_2$-saturated 0.1 M KOH. The results show that the average n of FeNSC-850 from 0 to 0.8 V (vs. RHE) is calculated to be 3.97, which is higher than that of MIL-88-850 (average $n$ = 3.26) and very close to the value (3.92) calculated from K-L plots. The yield of $HO_2^-$ intermediate is about 41% for MIL-88-850 at 0.47 V (vs. RHE), suggesting that there exists a two-electron ORR pathway from $O_2$ to $HO_2^-$, mainly due to the low overall content of N and less Fe-based active sites. The $HO_2^-$ yield of FeNSC-850 (<2.6%) is comparable to that of 20 wt% Pt/C ($HO_2^-$% < 1.1%) at 0.47 V (vs. RHE). Together with the electron transfer number, which is comparable to the commercial Pt/C electrocatalyst (average $n$ = 3.96), it indicates that FeNSC-850 displays a four-electron ORR pathway in 0.1 M KOH solution.

To further evaluate the durability of FeNSC-850 and 20 wt% Pt/C, cyclic voltammetry (CV) analysis was employed by scanning 5000 CV cycles from 0.5 to 1.1 V (vs. RHE) at 100 mV $s^{-1}$. As shown in Figure 6a, the $E_{1/2}$ of FeNSC-850 decreases by only 15 mV, which is less than that of 20 wt% Pt/C ($E_{1/2}$ loss = 63 mV). In comparison with others' work summarized in Table 2, our as-prepared NNMEs exhibit comparable or even superior ORR performances considering their catalytic ORR performances and stability. Furthermore, beneficial for practical fuel cell applications, chronoamperometric (I-t) measurements were conducted to assess the methanol tolerance of FeNSC-850 and the commercial Pt/C by the addition of 10 vol% methanol in $O_2$-saturated 0.1 M KOH solution at around 400 s. The normalized current density in Figure 6b suggests that unlike Pt/C, FeNSC-850 displays outstanding methanol tolerance, whereas Pt/C presents a substantial decline at the point of injection of methanol in $O_2$-saturated 0.1 M KOH media at around 400s.

To the best of our knowledge, the possible reasons for the efficient ORR catalytic activity of FeNSC-850 can be described as follows: (1) The large surface area with abundant microporous architecture guarantees the efficient mass transportation during the ORR process. (2) The higher content of graphitic N is more favorable in the N, S-doped carbon materials [45]. (3) The S-doping may change the microenvironment of iron-based active species during the pyrolysis step. (4) The S-S bonds in the main active sites of $FeS_x$ could provide electrons to the iron nitrides to promote the adsorption and reduction of $O_2$ through four electron ORR pathway [30].

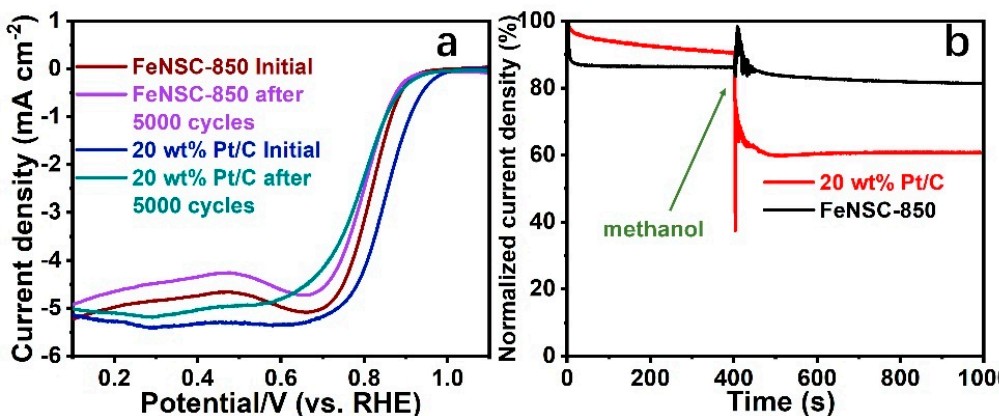

**Figure 6.** (**a**) LSV curves of FeNSC-850 at 1600 rpm with a scanning rate of 5 mV s$^{-1}$ before and after 5000 cycles in O$_2$-saturated 0.1 M KOH solution, (**b**) I-t response of FeNSC-850 and commercial Pt/C with the injection of methanol at ~400s in O$_2$-saturated 0.1 M KOH electrolyte.

**Table 2.** Comparison of ORR performances of FeNSC-850 with some representative reports in 0.1 M KOH solution.

| Sample | $E_{1/2}$ (V vs. RHE) | n [a] /RRDE | n [b] /K-L | HO$_2^-$% /RRDE | Stability | Ref. |
|---|---|---|---|---|---|---|
| Fe/S/N/C-2 | 0.85 | 3.93–3.98 | ~4 | 0.92–3.91 (0.1–0.9 V) | $\Delta E_{1/2}$ = 21 mV (10000 CV cycles) | [46] |
| FeNSC800 | 0.78 | 3.77–3.90 | 3.56–3.87 | <11.4 | No obvious decay (3000 CV cycles) | [47] |
| Fe-N/S-C | 0.87 | - | 3.9 | <5 | No obvious decay (2000 CV cycles) | [48] |
| Fe$_{1-x}$S@NSC-24 | 0.70 | - | 3.82–4.00 | - | No obvious decay (4000 CV cycles) | [49] |
| Co@NSC-1000 | 0.83 | - | 3.7 | - | $\Delta E_{1/2}$ = 18 mV (5000 CV cycles) | [50] |
| Fe,S/NGC-900 | 0.83 | 3.96 | 3.98−4.00 | <2 | No obvious decay (20000 s I-t) | [36] |
| Co@NPC@FeS$_2$-0.5 | 0.85 | 3.86–3.88 | ~4 | 5.1−5.6 (0–0.8 V) | $\Delta E_{1/2}$ = 13 mV (10000 CV cycles) | [30] |
| FeS/N,S:CNT–GR | 0.83 | - | ~4 | - | $\Delta E_{1/2}$ = 6 mV (6000 CV cycles) | [51] |
| FeNSC-850 | 0.82 | 3.97 | 3.92 | <2.6 | $\Delta E_{1/2}$ = 15 mV (5000 CV cycles) | This work |

[a] n is determined by the data from RRDE. [b] n is determined by the data from K-L.

## 3. Materials and Methods

### 3.1. Chemical Materials

Fe(NO$_3$)$_3$·9H$_2$O and NH$_4$Fe(SO$_4$)$_2$·12H$_2$O were purchased from Nanjing Chemical Reagent Co., Ltd. (Nanjing, China) and Shanghai Hushi Laboratorial Equipment Co., Ltd. (Shanghai, China). 2-Amino terephthalic acid was purchased from Shanghai Aladdin Biochemical Technology Co., Ltd. (Shanghai, China) 2-Amino-3-mercaptopropanoic acid (L-Cysteine) was obtained from Bide Pharmatech (Shanghai, China) Ltd. N, N-dimethyl-formamid (DMF) was purchased from Tianjin Dongli Tianda Chemical Reagent Plant (Tianjin, China). Methanol and ethanol were purchased from Shenyang East China Reagent Plant (Shenyang, China) and China National Pharmaceutical Industry Co., Ltd. (Beijing, China) All chemical reagents in this work were used without any further purification.

Ultrapure water with the resistance of 18.2 MΩ cm (at 25 °C) was used for all experiments and electrochemical properties.

### 3.2. Synthesis of MIL-88-850

MIL-88 with some modifications was prepared in light of the work of a report [52]. Typically, 0.50 g of 2-aminoterephthalate was firstly dissolved in 10 mL N, N-dimethylformamide with a vigorous stirring. Secondly, 1.12 g $Fe(NO_3)_3 \cdot 9H_2O$ was added in the above solution with further stirring for 2 h. Afterwards, the mixture was transferred into a Teflon-lined stainless steel autoclave and heat-treated at 160 °C for 6 h. Subsequently, the mixture was centrifuged and washed with DMF and methanol. The resulting sample was obtained after drying in a vacuum at 60 °C for 24 h. The obtained MIL-88 was directly pyrolyzed at 850 °C for 2 h with a ramp rate of 10 °C min$^{-1}$ in an Ar atmosphere. Afterwards, the obtained sample was acid-leached with 0.5 M $H_2SO_4$ overnight and washed with water until the filtrate became neutral. Finally, the sample was dried and denoted as MIL-88-850.

### 3.3. Synthesis of FeNSC-850

The obtained 1 g MIL-88 was dispersed into a 50 mL solution containing 25 mL water and 25 mL ethanol. Then, the certain mass ratio (1:1) of chitosan (500 mg) and L-Cy (500 mg), and 48 mg $NH_4Fe(SO_4)_2 \cdot 12H_2O$, was further added into the above solution with stirring at 60 °C for 3 h, followed by drying in the oven at 60 °C overnight. The obtained mixture was then placed closely at the middle of the furnace and heated at 850 °C for 2 h with a ramp rate of 10 °C min$^{-1}$ in an Ar atmosphere. Finally, the samples were leached with 0.5 M $H_2SO_4$ overnight and washed with water until the filtrate reached neutrality. Finally, the sample was dried and marked as FeNSC-850. For comparison, reference samples with a certain mass ratio of chitosan and L-Cy (1:0 and 0:1) and the sample without the addition of Fe salt were synthesized under the same conditions.

### 3.4. Characterization

The specific surface area and the corresponding pore size distribution of as-synthesis materials were calculated by means of the Brunauer–Emmett–Teller theory and were determined on the adsorption isotherm by Barrett–Joyner–Halenda (BJH) models. TEM and SEM were carried out on TECNAI G2F30 (FEI Company, USA) and S-4800 (Hitachi, Tokyo, Japan), respectively. XRD was conducted by X'pert Pro-1 (PANAlytical, Almelo, Netherlands) from 10 to 90°. The XPS of samples was conducted on Escalab Xi+ (Thermo Fisher Scientific, Winsford, UK). The Mössbauer spectra were collected at the ambient environment using the $^{57}$Co radioactive source and Topologic 500 A spectrometer. The Fe content of the electrocatalysts was measured by inductively coupled plasma optical emission spectroscopy (ICP-OES) (ICPS-8100, Shimadzu, Japan). Raman spectra were recorded on a Raman spectroscopy (Bruker Optics Senterra, Leipzig, Germany) with 532 nm excitation laser.

### 3.5. Electrochemical Measurements

The electrochemical properties, including the cyclic voltammetry (CV), the rotating disk electrode (RDE), the rotating ring-disk electrode (RRDE), and the I-t tests, were evaluated with a three-electrode system using CHI 760E (Chenhua, Shanghai, China) and VSP-300 (BioLogic, Paris, France). In this three-electrode system, the Hg/HgO and a graphite rod were used as the reference and counter electrode, respectively. In this paper, all the potential values were calibrated to the reversible hydrogen electrode (RHE) as $E_{RHE} = E_{Hg/HgO} + 0.884$ V. The calibration method for RHE follows the reported work [53]. The as-prepared electrocatalyst film-coated RDE and RRDE were utilized as the working electrodes. The electrocatalyst ink was prepared by dispersing 7.7 mg electrocatalysts in a solvent mixed with ethanol (900 µL), nano-pure water (100 µL), and Nafion (4.62 µL, 5 wt%). Afterwards, the electrocatalyst ink (20.4 µL) was dropped onto the polished glassy carbon (GC) on RDE. By air-drying at room temperature, the obtained electrocatalyst loading

on GC was calculated to be 0.8 mg cm$^{-2}$. In contrast, commercial Pt/C was prepared as 1 mg mL$^{-1}$, and 10 and 20 μL were further coated on the GCs to obtain 10 and 20 μg$_{Pt}$ cm$^{-2}$.

First of all, CV measurements were assessed at the potential of 0 to 1.1 V (vs. RHE) at a scanning rate of 100 mV s$^{-1}$, after which the surface of electrocatalyst could be further activated on GC. The ORR performances were conducted at 1600 rpm using a scanning rate of 5 mV s$^{-1}$. Both the RDE and RRDE tests were conducted in O$_2$ and N$_2$-saturated alkaline solution. Moreover, the RRDE technique was further determined at a consistent potential of 1.1 V (vs. RHE). To further discuss the electron transfer number, the Equations (1)–(4) were used as follows:

$$\frac{1}{J} = \frac{1}{J_K} + \frac{1}{B\omega^{\frac{1}{2}}} \tag{1}$$

$$B = 0.62 n F D_o^{2/3} v^{-1/6} C_o \tag{2}$$

In the Equation (1), J and J$_K$ are the current density and kinetic current density. In the Equation (2), n is the electron transfer number. $C_o$ (concentration of dissolved O$_2$) is $1.2 \times 10^{-6}$ mol cm$^{-3}$. $D_o$ (diffusion coefficient of O$_2$) is $1.9 \times 10^{-5}$ cm$^2$ s$^{-1}$. $v$ is 0.01 cm$^2$ s$^{-1}$. $F$ (Faraday constant) is 96,485 C mol$^{-1}$.

$$n = 4 \times \frac{I_D}{\frac{I_R}{N} + I_D} \tag{3}$$

$$HO_2^- \% = 200 \times \frac{\frac{I_R}{N}}{\frac{I_R}{N} + I_D} \tag{4}$$

$I_D$ and $I_R$ are the disk current and ring current, respectively. $N$ is the current collection efficiency of 37%.

Normally, the half-wave potential (E$_{1/2}$) and onset potential (E$_{onset}$) are used to evaluate the ORR performances of the obtained electrocatalysts. Here in this work, all E$_{1/2}$ and E$_{onset}$ were detected by the methods reported in our previous work [54].

Moreover, durability tests were conducted for 5000 CV cycles in the potential of 0.5 to 1.1 V vs. RHE at 100 mV s$^{-1}$ in O$_2$-saturated 0.1 M KOH media. The LSV curves were further conducted on the FeNSC-850 and Pt/C before and after 5000 cycles. Moreover, the I-t test of FeNSC-850 and commercial Pt/C were further assessed in the O$_2$-saturated 0.1 M KOH solution with the addition of methanol at around 400 s.

## 4. Conclusions

To summarize, we reported a synthetic route based on an MOF (MIL-88) template with property modification by N-/N, S-containing ligands and iron salt. After pyrolysis at 850 °C and acid leaching, the resulting FeSNC-850 owns a tremendous improvement in ORR performance. We also discovered that this improvement profits from the synergic effect between the supplemental FeS$_x$ sites generated by this method and the Fe-based active sites. The catalytic activity of FeNSC-850 towards ORR in alkaline solution is comparable to that of Pt/C, but the former also presents more durable properties as well as functional stability in a methanol environment. This work investigates FeS$_x$ as the main active site within multiple Fe-based active sites in ORR electrocatalysts.

**Supplementary Materials:** The following are available online at https://www.mdpi.com/article/10.3390/catal12080806/s1, Figure S1: the SEM images of (a) and (b) MIL-88-850; Figure S2: the EDX of FeNSC-850m; Figure S3: the XRD patterns of (a) MIL-88-850, (b) FeNSC-750, (c) FeNSC-800, and (d) FeNSC-900; Figure S4: the Raman spectra of (a) FeNSC-850 and (b) MIL-88-850; Figure S5: the survey spectra of (a) MIL-88-850 and (b) FeNSC-850; and Figure S6: the N 1s XPS spectra of (a) MIL-88-850, (b) FeNSC-750, (c) FeNSC-800, and (d) FeNSC-900.

**Author Contributions:** Investigation, methodology, and formal analysis, Y.L. and Y.X. (Yinghao Xu); methodology, H.Z.; characterization, Y.X. (Yinghao Xu) and J.Z.; characterization, H.W.; morphology, L.C.; conceptualization, writing—original draft preparation, and funding acquisition, Y.X. (Yan

Xie); supervision, writing—review and editing, and funding acquisition, L.X.; and supervision, writing—review, and editing, J.H. All authors have read and agreed to the published version of the manuscript.

**Funding:** This research was funded by the National Natural Science Foundation of China, No. 21606219, and Natural Science Foundation of Inner Mongolia, No. 2021LHMS02006.

**Data Availability Statement:** All data that support the findings of this study are included within the article (and any Supplementary Files).

**Acknowledgments:** We gratefully acknowledge support from Junhu Wang, the Mössbauer Effect Data Centre, and the Dalian Institute of Chemical Physics for the discussion on the Mössbauer spectra.

**Conflicts of Interest:** The authors declare no conflict of interest.

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
