# Peer review of "MIL-88-Derived N and S Co-Doped Carbon Materials with Supplemental FeSx to Enhance the Oxygen Reduction Reaction Performance"

_catalysts, doi:10.3390/catal12080806_

Round 1
Reviewer 1 Report
Review on
“MIL-88-Derived N and S Co-Doped Carbon Materials with Supplemental FeSx to Enhance the Oxygen Reduction Reaction Performance”
Liu et al. present a study on the beneficial effect of introducing Sulphur from L-Cysteine into a FeNC catalyst pyrolyzed from MIL-88 at 850 °C. The authors performed several characterization techniques revealing convincing results. The work is overall clearly structured and has an appropriate length. However, the language has to be improved in some sentences.
In general, I would appreciate a discussion on the advantages and disadvantages of the use of a second heteroatom for FeNC catalysts especially regarding the application in fuel cells. Furthermore, the authors should report the Fe-content of MIL-88 and discuss it together with the Fe content of the synthesized materials. Additionally, the composition of the catalyst in terms of Fe, C, N, S content (at%) should be presented. The results of the Fe 2p XP spectra should be discussed more carefully, because the spectrum is quite noisy with only low intensity and the fitting therefor prone to misinterpretation. In the electrochemistry section, the authors should explain their choice of taking ‑2.5 mA cm-2 for comparison of catalysts and the potential range of the durability test. How do they refer to fuel cell application?
In the experimental part, the authors should describe their approach for determining Eonset and E1/2.
· Line 74: I am not quite sure if Fe carbides are ORR active. The authors should check the literature on that.
· Line 78/79 Scheme1: Please add the synthesis condition as temperature, solvent, duration…
· Line 86/87: Is it really possible to see the different pores or pore sizes from the SEM image?
· Line 92/93: I would be rather careful to claim that the formed Fe-S species are encapsulated by graphitic carbon at that point. You do not know if the carbon is graphitic from these images.
· Line 97: Why nanoparticles?
· Line 101/102: I am not sure if XRD is the right method to claim the amount of graphitic part of a material. Are the peaks normalized? And what about the broad underlying peak (at around 25 ° 2Theta) especially in FeNSC-850? Where does it stem from? The degree of graphitization should rather be investigated by methods like Raman spectroscopy.
· Line 123/124: The authors should refer to literature regarding the location of active Fe sites within different pores.
· Line 182: Where to find the named “FeN4” in Fig. 4b?
· Line 188 Figure 4b: The legend contains two FeSx? How are they different?
· Line 246 Figure 5a/b: Please use consistent colors for the presented samples.
· Lines 276-279: What is the potential of the MeOH experiment?
Author Response
Comment 1:
- In general, I would appreciate a discussion on the advantages and disadvantages of the use of a second heteroatom for FeNC catalysts especially regarding the application in fuel cells.
Response: Thank you for your positive suggestions. We have added the advantages of FeNSC in the introduction and cited more references, like Ref. [10].
- Furthermore, the authors should report the Fe-content of MIL-88 and discuss it together with the Fe content of the synthesized materials.
Response: Thank you for your comments. The Fe content of MIL-88-850 is determined by ICP as 9.34 wt%. And the Fe content of FeNSC-850 is determined by ICP as 3.37 wt%. Obviously, the modified electrocatalysts lost some Fe content during the high-temperature pyrolysis and acid-leaching. As shown in the SEM images in Figure 1 and S1, the structure of MIL-88-850 is maintained while the some of the structure of FeNSC-850 collapsed, thus some Fe-based NPs were leached out. As a result, the ICP content of Fe in our synthesized electrocatalyst of FeNSC-850 is lower than that of MIL-88-850.
- Additionally, the composition of the catalyst in terms of Fe, C, N, S content (at%) should be presented. The results of the Fe 2p XP spectra should be discussed more carefully, because the spectrum is quite noisy with only low intensity and the fitting therefor prone to misinterpretation.
Response: Thank you very much for your positive suggestions. We have added the full scanning of XPS of MIL-88-850 and FeNSC-850 in Figure S5 in the SI. Maybe most Fe based NPs are encapsulated within thicker carbon layers (> 10 nm) that cannot be detected beyond its test line. Thus, there is not clear signals of Fe in the XPS.
- In the electrochemistry section, the authors should explain their choice of taking ‑2.5 mA cm-2 for comparison of catalysts and the potential range of the durability test. How do they refer to fuel cell application?
Response: Thank you for your comments. The limiting current for oxygen diffusion should be flat. While for NNMEs, the limiting current of some electrocatalysts are not like that, thus, for comparison purposes, we choose a representative current density of ‑2.5 mA cm-2 for further comparison of electrocatalysts towards ORR performances. On the other hand, the value of ‑2.5 mA cm-2 is very close to but not the E1/2 value of FeNSC-850
- In the experimental part, the authors should describe their approach for determining Eonset and E1/2.
Response: Thank you for your comments. E1/2 and Eonset are determined like our previous work. And the Ref. [28] is cited in the revised paper.
- Line 74: I am not quite sure if Fe carbides are ORR active. The authors should check the literature on that.
Response: Thank you for your comments. The Fe carbides are one of the active sites for ORR. We have cited the corresponding references (Ref. 24) in the revised manuscript. While, Fe carbides display less competitive ORR performances when FeNx exists in the ORR progress.
Line 78/79 Scheme1: Please add the synthesis condition as temperature, solvent, duration…
Response: Thank you for your positive suggestion. We have modified the Scheme 1 in the revised manuscript.
- Line 86/87: Is it really possible to see the different pores or pore sizes from the SEM image?
Response: Thank you for your comments. We have changed the SEM image in the paper and this time the pores can be observable in the SEM image in Figure 1b.
- Line 92/93: I would be rather careful to claim that the formed Fe-S species are encapsulated by graphitic carbon at that point. You do not know if the carbon is graphitic from these images.
Response: Thank you for your positive suggestion. We have revised “the graphitic carbon layers” as “carbon matrixes”.
- Line 97: Why nanoparticles?
Response: Thank you for your positive suggestion. We have removed the “nanoparticles (NPs)” in the sentence.
- Line 101/102: I am not sure if XRD is the right method to claim the amount of graphitic part of a material. Are the peaks normalized? And what about the broad underlying peak (at around 25 ° 2Theta) especially in FeNSC-850? Where does it stem from? The degree of graphitization should rather be investigated by methods like Raman spectroscopy.
Response: Thank you for your comments. The peaks in XRD are not normalized. The broad peak around 25o can be ascribed to the characteristic carbon (002). And the sharp peak around 26o can be ascribed to the Fe3C (002) (PDF #03-0411). We have rewritten it in the paper. Moreover, we have added the Raman spectroscopy of electrocatalysts in Figure S4. More descriptions were added in the paper for Raman analysis on electrocatalysts.
- Line 123/124: The authors should refer to literature regarding the location of active Fe sites within different pores.
Response: Thank you for your comments. We have added the description in the revised manuscript and cited the references.
- Line 182: Where to find the named “FeN4” in Fig. 4b?
Response: Thank you for your comments. We have revised it in Figure 4b, in which FeN4 can be found.
- Line 188 Figure 4b: The legend contains two FeSx? How are they different?
Response: Thank you for your comments. Regarding to the literature, the FeSx can be ascribed as the main components as shown in Table 1. Thus, FeSx is marked by a joint name, which may be ascribed as hexagonal pyrrhotite and monoclinic pyrrhotite. But in most paper, they can be generally denoted as FeSx. And we have cited the Ref.44 in the revised manuscript for that.
- Line 246 Figure 5a/b: Please use consistent colors for the presented samples.
Response: Thank you for your suggestions. We have revised it in Figure 5a/b with the consistent colors to present the samples.
- Lines 276-279: What is the potential of the MeOH experiment?
Response: Thank you for your suggestions. Response: Thank you for your suggestions. As reported, Fe– and Co–N/C catalysts are more active than Pt/C in the presence of methanol. This methanol tolerance is especially useful in a direct methanol fuel cell (DMFC), in which methanol crossing over the membrane from anode to cathode is one of the major barriers for further technology development and commercialization. The methanol that crosses over can react directly with cathodic O2 catalyzed by the Pt-based catalyst, depressing the O2 surface concentration of oxidant O2 and reducing the fuel efficiency. If Fe– and Co–N/C catalysts are used as DMFC cathode catalysts, the methanol that crosses over will not react with O2 because the catalyst has no ability to catalyze the reaction. Therefore, developing more selective cathode catalysts and/or methanol-tolerant catalysts for O2 reduction is very important in DMFC development. Due to their lack of or low activity for methanol oxidation, M–N/C catalysts have attracted enormous attention as promising catalyst materials. (Electrochimica Acta 53 (2008) 4937–4951)
Reviewer 2 Report
In this work, the authors prepared N, S-doping, and Fe-based carbon materials via pyrolyzing a metal-organic framework (MIL-88) with the addition of N-/N, S-containing ligands (chitosan and L-Cysteine) in the case of iron salt. The resulting electrocatalyst heat-treated at 850 (FeNSC-850) displays comparatively higher ORR performances in alkaline solution. Some of the comments areas below:
1. The abstract of the manuscript needs to rewrite with more details about the performance.
2. What is the novelty of this work, it should be elaborated in the introduction section with the required explanation.
3. Section 3 materials and method should be placed before section 2 results and discussion.
4. Where are the related characterization (SEM images and XRD ) of the MIL-88, MIL-88-800, FeNSC-750, FeNSC-800, and FeNSC-900. It should be added to the supporting information, making a separate file.
5. The EDX related to Fig. 1 elemental mapping should be added to supporting information.
6. The authors can use the following work for the data presentation and explanation references with the proper citation in the introduction section:10.1016/j.jcis.2022.03.104 and 10.1016/j.carbon.2021.04.028
7. As shown in Figure 3 (c), the different % of different types of N in the bar diagram, authors need to add comparative N 1S XPS spectra in supporting information.
8. What is the reason behind the lower % of N content in FeNSC-800 than in FeNSC-750 and FeNSC-850.
9. As shown in Figure 5 (a), the authors should add the individual ORR LSV of MIL-88-850, FeNSC-750, FeNSC-800, and FeNSC-900 at different rpm in supporting information as presented in Figure 5 (d).
10. As presented in Figure 6 (a) a Table 2, FeNSC-850 after 5000 cycles but there is no data of 5000 cycles of stability test data? It needs to be shown in the main manuscript.
11. THe STEM image shown in Figure 1 (e) is not clear. It needs to be retaken.
Author Response
- The abstract of the manuscript needs to rewrite with more details about the performance.
Response: Thank you for your suggestions. We have revised the abstract and added more details in it.
- What is the novelty of this work, it should be elaborated in the introduction section with the required explanation.
Response: Thank you for your suggestions. We have modified the introduction in the revised paper.
- Section 3 materials and method should be placed before section 2 results and discussion.
Response: Thank you for your suggestions. We have removed the experiment section before results and discussion.
- Where are the related characterization (SEM images and XRD ) of the MIL-88, MIL-88-800, FeNSC-750, FeNSC-800, and FeNSC-900. It should be added to the supporting information, making a separate file.
Response: Thank you for your suggestions. The SEM images of MIL-88-850 have been removed to Figure S1 in SI. And XRD of MIL-88-850, FeNSC-750, FeNSC-800 and FeNSC-900 have been added in Figure S3 in SI.
- The EDX related to Fig. 1 elemental mapping should be added to supporting information.
Response: Thank you for your suggestions. We have added the EDX of FeNSC-850 in Figure S2.
- The authors can use the following work for the data presentation and explanation references with the proper citation in the introduction section:10.1016/j.jcis.2022.03.104 and 10.1016/j.carbon.2021.04.028
Response: Thank you for your suggestions. We have cited these two references in the introduction section as Ref. [20] and [21].
- As shown in Figure 3 (c), the different % of different types of N in the bar diagram, authors need to add comparative N 1S XPS spectra in supporting information.
Response: Thank you for your suggestions. We have added the comparative N1s XPS spectra in Figure S6.
- What is the reason behind the lower % of N content in FeNSC-800 than in FeNSC-750 and FeNSC-850.
Response: Thank you for your comments. We redetected the XPS N1s of FeNSC-800 and FeNSC-750. But we still do not get back the XPS data of them due to the instrument issues. Thus, we would modify the data in the next revision or in the proof step.
- As shown in Figure 5 (a), the authors should add the individual ORR LSV of MIL-88-850, FeNSC-750, FeNSC-800, and FeNSC-900 at different rpm in supporting information as presented in Figure 5 (d).
Response: Thank you for your comments. We did not present different ORR LSV of other samples due to the fact that FeNSC-850 displays better ORR performances than those of MIL-88-850, FeNSC-750, FeNSC-800, and FeNSC-900. The purpose of different ORR LSV of the electrocatalyst is for the calculation on transfer number of reduction of O2 to compare the performance with the commercial Pt/C. Thus, we choose to calculate the representative and the best electrocatalyst of FeNSC-850 in our work to show that our synthesized electrocatalyst exhibits very close transfer number (~4) as to the commercial Pt/C in 0.1 M KOH solution.
- As presented in Figure 6 (a) a Table 2, FeNSC-850 after 5000 cycles but there is no data of 5000 cycles of stability test data? It needs to be shown in the main manuscript.
Response: Thank you for your comments. The Figure R1 below shows that 5000 CV cycles of FeNSC-850 for the durability test, but in most cases and reported work, the 5000 CV cycles data are not shown in the main paper. Commonly, the LSV curves of electrocatalysts before and after 5000 scanning are shown in the paper as the data in Figure 6a.
Figure R1. Please see the word.
Figure R1. The 5000 cycles of FeNSC-850 in 0.1 M KOH solution.
- The STEM image shown in Figure 1 (e) is not clear. It needs to be retaken.
Response: Thank you for your suggestions. We have modified the STEM image in Figure 1c.

Round 2
Reviewer 2 Report
Accept in present form.